# *Bacteroides thetaiotaomicron* Fosters the Growth of Butyrate-Producing *Anaerostipes caccae* in the Presence of Lactose and Total Human Milk Carbohydrates

**DOI:** 10.3390/microorganisms8101513

**Published:** 2020-10-01

**Authors:** Loo Wee Chia, Marko Mank, Bernadet Blijenberg, Steven Aalvink, Roger S. Bongers, Bernd Stahl, Jan Knol, Clara Belzer

**Affiliations:** 1Laboratory of Microbiology, Wageningen University and Research, 6708 WE Wageningen, The Netherlands; loowee.chia@frieslandcampina.com (L.W.C.); steven.aalvink@wur.nl (S.A.); Jan.Knol@danone.com (J.K.); 2Danone Nutricia Research, 3584 CT Utrecht, The Netherlands; Marko.MANK@danone.com (M.M.); Bernadet.Blijenberg@danone.com (B.B.); Roger.BONGERS@danone.com (R.S.B.); Bernd.STAHL@danone.com (B.S.); 3Department of Chemical Biology & Drug Discovery, Utrecht Institute for Pharmaceutical Sciences, Utrecht University, 3584 CT Utrecht, The Netherlands

**Keywords:** *Bifidobacteria*, cross-feeding, d-lactate, human milk oligosaccharides, lactose

## Abstract

The development of infant gut microbiota is strongly influenced by nutrition. Human milk oligosaccharides (HMOSs) in breast milk selectively promote the growth of glycan-degrading microbes, which lays the basis of the microbial network. In this study, we investigated the trophic interaction between *Bacteroides thetaiotaomicron* and the butyrate-producing *Anaerostipes caccae* in the presence of early-life carbohydrates. Anaerobic bioreactors were set up to study the monocultures of *B. thetaiotaomicron* and the co-cultures of *B. thetaiotaomicron* with *A. caccae* in minimal media supplemented with lactose or a total human milk carbohydrate fraction. Bacterial growth (qPCR), metabolites (HPLC), and HMOS utilization (LC-ESI-MS^2^) were monitored. *B. thetaiotaomicron* displayed potent glycan catabolic capability with differential preference in degrading specific low molecular weight HMOSs, including the neutral trioses (2′-FL and 3-FL), neutral tetraoses (DFL, LNT, LNnT), neutral pentaoses (LNFP I, II, III, V), and acidic trioses (3′-SL and 6′-SL). In contrast, *A. caccae* was not able to utilize lactose and HMOSs. However, the signature metabolite of *A. caccae*, butyrate, was detected in co-culture with *B. thetaiotaomicron*. As such, *A. caccae* cross-fed on *B. thetaiotaomicron*-derived monosaccharides, acetate, and d-lactate for growth and concomitant butyrate production. This study provides a proof of concept that *B. thetaiotaomicron* could drive the butyrogenic metabolic network in the infant gut.

## 1. Introduction

The establishment of infant gut microbiota is a dynamic process with successive colonization of the functionally distinct microbial groups [1,2]. The infant gut microbiota displays high temporal and inter-individual variation that is influenced by factors including genetics, the mode of delivery, hospitalization, the use of antibiotics, and early nutrition [3]. Even though a “healthy” infant gut microbiota has not been well defined, it is associated with desirable criteria such as: full-term delivery via natural birth, no antibiotic use, and breast feeding. Breast feeding selectively promotes the growth of bacteria capable of utilizing human milk oligosaccharides (HMOSs). HMOSs are non-digestible carbohydrates with prebiotic functions that selectively stimulate the growth and activity of specific microbes in the infant gut [4]. The primary HMOS degraders, i.e., *Bifidobacterium* spp., are the dominant taxonomic group in the gut of healthy infants and, to a lesser extent, are *Bacteroides* spp. [1].

A trade-off between the abundance of *Bifidobacterium* spp. and *Bacteroides* spp. is often observed in infants [5]. Interestingly, infants delivered by caesarean section are deprived of *Bacteroides* spp. in the first days of life, likely due to the absence of exposure to members of this genus in the natural birth canal [1,6]. The abundance of *Bacteroides* spp. increases until the establishment of a stable adult gut microbiota that predominantly comprises Bacteroidetes and Firmicutes [2]. The common *Bacteroides* spp. found in the infant gut are *Bacteroides thetaiotaomicron*, *Bacteroides fragilis*, and *Bacteroides vulgatus* [1,6]. *B. thetaiotaomicron* is generally recognized as a symbiont that contributes to postnatal gut development and host physiology [7]. *B. thetaiotaomicron* is capable of foraging on host glycans, including mucin and HMOSs [8]. The distinct capability to utilize glycans using the archetypal starch utilization system (Sus) and the ability to sense and respond to the environmental cues lead to colonies of *B. thetaiotaomicron* in human gut [9,10].

The primary HMOS degraders could drive the microbial community in the infant gut via cross-feeding. As such, the intermediate breakdown products by one bacterial species serve as the substrate to support the growth of other microbes in the environment, resulting in indirect benefits for all species involved [11,12,13,14]. For instance, *Bifidobacterium* spp. degrade host-produced 2′-fucosyllactose (2′-FL, an abundant HMOS) and mucin glycans to support butyrate-producing bacteria [15,16]. Furthermore, the in vivo interaction between *B. thetaiotaomicron* and butyrate-producing bacteria in gnotobiotic mice led to an increase in intestinal butyrate [17,18]. However, to date, no study has addressed the interaction between *Bacteroides* spp. and butyrogen in early-life carbohydrates. We hypothesize that *Bacteroides* spp. could also drive the butyrogenic metabolic network in the infant gut.

This study investigates the trophic interaction between the early-life colonizers, i.e., *B. thetaiotaomicron* and butyrate-producing *Anaerostipes caccae* from the Lachnospiraceae family [1,19]. *A. caccae* is observed in the infant gut microbiota [1,20] and is identified as a key bacterium in regulating food allergies early in life [19]. Anaerobic bioreactor culturing of the bacteria on early-life carbohydrates, including lactose and total human milk (HM) carbohydrates, has been conducted. We demonstrated that *B. thetaiotaomicron* could support the growth of putative beneficial butyrate-producing bacteria using early-life carbohydrates.

## 2. Materials and Methods

### 2.1. Bacterial Strains and Growth Conditions

Bacterial pre-cultures were prepared by growing them overnight in anaerobic serum bottles sealed with butyl rubber stoppers at 37 °C with a gas phase of N_2_/CO_2_ (80/20 ratio) at 1.5 atm. Basal medium [21] containing 0.5% (w/v) tryptone (Oxoid, Basingstoke, UK) was used for culturing. For *Bacteroides thetaiotaomicron* DSM 2079 (VPI 5428), 30 mM of lactose (Oxoid, Basingstoke, UK) and 5 mg/L of hemin (Sigma-Aldrich, St. Louis, MO, USA) were supplemented; whereas for *Anaerostipes caccae* DSM 14662 (L1-92) [22], 30 mM of glucose (Sigma-Aldrich, St. Louis, MO, USA) was supplemented. Growth was measured by a spectrophotometer as optical density at 600 nm (OD600) (OD600 DiluPhotometerTM, IMPLEN, München, Germany).

### 2.2. Growth Substrates

Lactose (Oxoid, Basingstoke, UK) and total human milk (HM) carbohydrate fractions were tested as the growth substrates. The usage and analysis of pooled human milk samples described in this study were performed in accordance with ethical standards and guidelines as laid down in the Declaration of Helsinki. Ethical approval and the written consent of donors were given as stated in Thurl et al. [23]. For the preparation of total HM carbohydrate fractions, a total carbohydrate mineral fraction was derived from pooled human milk after protein depletion by ethanol precipitation and the removal of lipids by centrifugation, as described by Stahl et al. [24]. Deviating from this workflow, no anion exchange chromatography was used to further separate neutral from acidic oligosaccharides present in the resulting total carbohydrate mineral fraction. The total HM carbohydrate fraction contained approximately 10% HMOSs and 90% lactose, as estimated by gel permeation chromatography (GPC) [25].

### 2.3. Anaerobic Bioreactor

Fermentations were conducted in eight parallel mini-spinner bioreactors (DASGIP, Germany) with 100 mL filling volume at 37 °C and a stirring rate of 150 rpm. Culturing experiments were performed in autoclaved basal media [21] containing 0.5% (w/v) tryptone (Oxoid, Basingstoke, UK) supplemented with 8 g/L of 0.2 µM filter-sterilized lactose or total HM carbohydrate fraction. Experiments were performed with 1% (v/v) supplementation of 0.2 µM filter-sterilized 0.5 g/L hemin stock solution and B vitamin stock solution. The B vitamin stock solution contained 11 g/L CaCl_2_, 20 mg/L biotin, 200 mg/L nicotinamide, 100 mg/L p-aminobenzoic acid, 200 mg/L thiamine, 100 mg/L pantothenic acid, 500 mg/L pyridoxamine, 100 mg/L cyanocobalamin, and 100 mg/L riboflavin. An anaerobic condition was achieved by overnight purging of an anaerobic gas mixture that contained 5% CO_2_, 5% H_2_, and 90% N_2_. Overnight pre-cultures were inoculated at a starting OD600 of 0.05 for each bacterial strain. Online signals of pH values and oxygen levels were monitored by DASGIP Control software (DASGIP, Jülich, Germany). Cultures were maintained at pH 6.5 by dosing 2 M sodium hydroxide (NaOH).

### 2.4. Gel Permeation Chromatography (GPC)

Total HM carbohydrates were analyzed using GPC. Glycans were separated by the GPC stationary phase and eluted according to size and charge. Neutral mono-, di-, and oligosaccharides, and acidic oligosaccharides with different degrees of polymerization (DP), could be detected. HM carbohydrate solution was prepared by dissolving 0.2 g/mL of total HM carbohydrates in ultrapure water (Sartorius Arium Pro) containing 2% (v/v) 2-propanol at 37 °C. Five milliliters of 0.2 µM filter-sterilized HM carbohydrate solution were injected for each GPC run. The sample loop was cleaned by ultrapure water prior to analysis. Two connected Kronlab ECO50 columns (5 × 110 cm) packed with Toyopearl HW 40 (TOSOH BIOSCIENCE) were used. Milli-Q water was maintained at 50 °C using a heating bath (Lauda, RE 206) for column equilibration. Milli-Q water containing 2% (v/v) 2-propanol was used as the eluent. The flow rate of the eluent was set at 1.65 mL/min. Eluting glycans were monitored by refractive index detection (Shodex, RI-101). The resulting chromatograms were analyzed by using Chromeleon^®^ software (Thermo Scientific).

### 2.5. High-Performance Liquid Chromatography (HPLC)

For metabolite analysis, 1 mL of bacterial culture was centrifuged, and the supernatant was stored at −20 °C until HPLC analysis. Crotonate was used as the internal standard, and the external standards tested were lactose, glucose, galactose, N-acetylglucosamine (GlcNAc), fucose, malate, fumarate, succinate, citrate, formate, acetate, butyrate, isobutyrate, lactate, 1,2-propanediol, and propionate. Metabolites were measured with a Spectrasystem HPLC (Thermo Scientific, Breda, the Netherlands) equipped with a Hi-Plex-H column (Agilent, Amstelveen, the Netherlands) for the separation of organic acids and carbohydrates. A Hi-Plex-H column performed separation with diluted sulfuric acid on the basis of ion exchange–ligand exchange chromatography. Measurements were conducted at a column temperature of 45 °C with an eluent flow of 0.8 mL/min of 0.01 N sulfuric acid. Metabolites were detected by refractive index (Spectrasystem RI 150, Thermo, Breda, the Netherlands). The data were analyzed with Chromeleon^®^ software (Thermo Scientific).

### 2.6. HMOS Extraction

HMOSs were recovered from 1 mL aliquots of bacterial cultures. The internal standard, 1,5-α-l-arabinopentaose (Megazyme), was added, at a volume of 10 µL per sample to minimize pipetting error, to reach a final concentration of 0.01 mmol/L. The solution was diluted 1:1 with ultrapure water and centrifuged at 4000× *g* for 15 min at 4 °C. The supernatant was filtered through a 0.2 μM syringe filter followed by subsequent centrifugation with a pre-washed ultra-filter (Amicon Ultra 0.5 Ultracel Membrane 3 kDa device, Merck Milipore) at 14,000× *g* for 1 h at room temperature. Finally, the filtrate was vortexed and stored at −20 °C until further targeted liquid chromatography electrospray ionization tandem mass spectrometry (LC-ESI-MS^2^) analysis.

### 2.7. Targeted Liquid Chromatography Electrospray Ionization Tandem Mass Spectrometry (LC-ESI-MS^2^) Analysis

The identification and relative quantitation of HMOSs were accomplished by LC-ESI-MS^2^. This method allowed for the study of distinct HMOS structures differing in monosaccharide sequence, glycosidic linkage, or molecular conformation. Thereby, even isobaric HMOS isomers such as lacto-N-fucopentaose (LNFP) I, II, III, and V could be distinguished, as described by Mank et al. [26]. LC-ESI-MS^2^ analysis was performed on a 1200/1260 series HPLC stack (Agilent, Waldbronn, Germany) consisting of a solvent tray, degasser, binary pump, autosampler, and DAD detector coupled to a 3200 Qtrap mass spectrometer (ABSciex, Framingham, MA, USA). After HMOS clean up by 3 kDA ultrafiltraton and spiking with the internal standard 1,5-α-l-arabinopentaose (see above), 5 µL of HMOS ultrafiltration permeate were injected into the LC-MS system. Oligosaccharides were separated by means of a 2.1 × 30 mm Hypercarb porous graphitized carbon (PGC) column with a 2.1 × 10 mm PGC pre-column (Thermo Scientific, Waltham, MA, USA). HMOSs were separated using a 30 min water–methanol gradient after 2 min of pre-equilibration. Solvent A consisted of 0.3% (v/v) ammonium hydroxide solution (28–30%, Sigma-Aldrich, St. Louis, MO, USA) in water and solvent B of 0.3% (v/v) ammonium hydroxide solution in 95% (v/v) methanol. Eluent flow was 400 μL/min and the columns were kept at 45 °C. Pre-equilibration was performed using 97.5% solvent A. The gradient started with 97.5% solvent A for 0.5 min, decreased to 60% in 12.5 min, and decreased to 40% in 3 min, where it was kept for 4 min. In a next segment, solvent A decreased in 0.5 min to 2.5%, where it was kept for 3 min. In 0.5 min, solvent A increased to 97.5% for a re-equilibration of 6 min. Individual HMOS structures eluted from the column were infused into the mass spectrometer and analyzed qualitatively and quantitatively by multiple reaction monitoring (MRM) in negative ion mode. Specific MRM transitions for neutral HMOSs up to pentoses and acidic HMOSs up to trioses were employed. The spray voltage was −4500 V and the declustering potential and collision energy were optimized to the individual compounds measured. Each MRM transition was measured for 70 ms. The instrument was calibrated with polypropylene glycol according to the instructions of the manufacturer.

### 2.8. Quantitative Real-Time PCR (qPCR)

The abundances of *B. thetaiotaomicron* and *A. caccae* in co-culture were determined by quantitative real-time PCR. Bacterial cultures were harvested at 16,100× *g* for 10 min at 4 °C. DNA extractions were performed using a MasterPure™ Gram-Positive DNA Purification Kit. DNA concentrations were determined fluorometrically (Qubit dsDNA HS assay; Invitrogen) and adjusted to 1 ng/μL prior to use as the template in qPCR. Primers targeting the 16S rRNA gene of *B. thetaiotaomicron* (g-Bfra-F 5′-ATAGCCTTTCGAAAGRAAGAT-3′; g-Bfra-R 5′-CCAGTATCAACTGCAATTTTA-3′; 501 bp product [27]) and *A. caccae* (OFF2555 5′-GCGTAGGTGGCATGGTAAGT-3′; OFF2556 5′-CTGCACTCCAGCATGACAGT-3′; 83 bp product [28]) were used for quantification. Standard template DNA was prepared from the 16S rRNA gene of each bacterium by amplification with primers 27F (5′-AGAGTTTGATCCTGGCTCAG-3′) and 1492R (5′-GGTTACCTTGTTACGACTT-3′). Standard curves were prepared with nine standard concentrations of 10^0^ to 10^8^ gene copies/μL. PCRs were performed in triplicate with iQ SYBR Green Supermix (Bio-Rad) in a total volume of 10 μL with primers at 500 nM in 384-well plates sealed with optical sealing tape. Amplification was performed with an iCycler (Bio-Rad, Contra Costa County, CA, USA) with the following protocol: one cycle of 95 °C for 10 min; 40 cycles of 95 °C for 15 s, 55 °C for 20 s, and 72 °C for 30 s each; one cycle of 95 °C for 1 min, one cycle of 60 °C for 1 min, and a stepwise increase of the temperature from 60 to 95 °C (at 0.5 °C per 5 s) to obtain melt curve data. Data were analyzed using Bio-Rad CFX Manager 3.0.

### 2.9. Statistical Analysis

Statistics were performed using *t*-tests and corrected for multiple testing using false discovery rate (FDR) correction for multiple comparisons. The *p*-values < 0.05 were considered significant.

## 3. Results

### 3.1. B. thetaiotaomicron Supported the Growth of A. caccae in the Presence of Early-Life Carbohydrates

*B. thetaiotaomicron* in mono- and co-culture with *A. caccae* were cultured in an anaerobic bioreactor regulated at pH 6.5 to simulate the conditions in the gut of infants. Lactose or total human milk (HM) carbohydrates were tested in basal medium supplemented with hemin and B vitamins. The monoculture of *B. thetaiotaomicron* grew in both lactose and total HM with a continuous increase in cell density till 72 h of fermentation (OD_max_ = 4.07 ± 0.10 in lactose and OD_max_ = 3.58 ± 4.31 in total HM carbohydrates) (Figure 1). No growth was observed for *A. caccae* (OD_max_ = 0.04 ± 0.01 in lactose and OD_max_ = 0.05 ± 0.01 in total HM carbohydrates) (Appendix A). *B. thetaiotaomicron* in co-culture with *A. caccae* resulted in synergistic growth reaching a maximum cell density at 24 h (OD_max_ = 5.47 ± 0.43 in lactose and OD_max_ = 4.84 ± 0.26 in total HM carbohydrates). The growth profiles were reflected in the acidification of the cultures. A qPCR was performed to monitor the growth of each bacterial strain in the co-culture. Around 6 log of cells were inoculated for *B. thetaiotaomicron* and *A. caccae* and both of the strains decreased 10-fold in abundance in the first 5 h. Subsequently, *B. thetaiotaomicron* grew exponentially to 1.24 × 10^9^ copy number/mL in lactose and 1.09 × 10^9^ copy number/mL in total HM carbohydrates at 11 h, after which growth slowed down. *A. caccae* showed a similar trend with an increase of abundance to 8.56 × 10^6^ copy number/mL in lactose and 1.93 × 10^7^ copy number/mL in total HM carbohydrates at 11 h. In both substrates, *B. thetaiotaomicron* outnumbered *A. caccae* by 100-fold.

### 3.2. Cross-Feeding Between B. thetaiotaomicron and A. caccae Leads to Butyrate Production

The sugar consumption and short-chain fatty acid (SCFA) production were monitored over time (Figure 2). Similar changes in the composition of metabolites were observed for the fermentation of lactose and total HM carbohydrates which consisted of approximately 10% HMOSs and 90% of lactose [25]. *B. thetaiotaomicron* monoculture showed limited substrate catabolism as a low amount of lactose (4.24 ± 0.73 mM in lactose and 1.15 ± 0.83 mM in total HM carbohydrates) was still detected after 72 h of fermentation. A low amount (around 1 mM) of monosaccharides, including glucose and galactose, were detected in the supernatant throughout the course of fermentation. *B. thetaiotaomicron* produced acetate, propionate, succinate, and lactate, as well as a low amount of malate from lactose and total HM carbohydrate fermentation. The co-culture of *B. thetaiotaomicron* with *A. caccae* showed a rapid consumption of lactose with complete depletion within 24 h. Up to 3 mM of glucose and galactose were detected in the supernatant of co-cultures after the first 12 h of fermentation. The major metabolites in the co-cultures were propionate, succinate, acetate, butyrate, and formate. Butyrate, the signature product of *A. caccae*, was produced at 11.85 ± 0.32 mM in lactose and 12.23 ± 1.80 mM in total HM carbohydrates. In contrast to the monoculture, several additional changes were observed in the metabolites of the co-culture, including no detection of lactate, a decrease in acetate and malate after 24 h, and the production of formate (5.70 ± 0.64 mM in lactose and 4.32 ± 0.65 mM in total HM carbohydrates).

### 3.3. Differential Utilization of HMOS Structures

The bacterial ferments of total HM carbohydrates were analyzed for HMOS-specific sugars and low molecular weight HMOS structures to investigate the glycan degradation capability of the bacteria. Both monitored HMOS-specific sugars, *N*-acetylglucosamine (GlcNAc) and fucose, were below the detection limit of 0.5 mM throughout the fermentation. The glycoprofiling analysis (Figure 3) showed that there was no HMOS degradation in the *A. caccae* monoculture. Differential kinetics of degradation were observed in the *B. thetaiotaomicron* monoculture for specific HMOS structures, with noticeable utilization of 2′-FL, 3-FL, 3′-SL, and 6′-SL at 24 h. Interestingly, at 24 h, an increase in the relative abundance of LNT, LNnT, LNFP I, and LNFP V was observed. Partial degradation of all quantified low molecular weight HMOSs, including neutral trioses (2′-FL and 3-FL), neutral tetraoses (DFL, LNT, LNnT), neutral pentaoses (LNFP I, II, III, V), and acidic trioses (3′-SL and 6′-SL), was observed in the *B. thetaiotaomicron* monoculture at 48 h. A more rapid overall HMOS degradation was observed in the *B. thetaiotaomicron* co-culture with *A. caccae* compared to the *B. thetaiotaomicron* monocultures. In the co-cultures, 2′-FL and LNFP III were completely degraded within 24 h, and most of the low molecular weight HMOSs were diminished within 48 h of fermentation.

## 4. Discussion

Bacterial cross-feeding on the non-digestible dietary components drives microbial network formation in the infant gut. The intricate relationship among key functional groups is vital to maintain health. Dysbiosis of infant gut microbiota could result in short-term consequences, such as intestinal discomfort and colic [29,30], as well as atopic and metabolic syndromes that compromise life-long health [31,32,33,34]. In this study, we investigated the role of early-life carbohydrates, i.e., lactose and HMOSs, in driving the butyrogenic microbial interaction.

The major microbial-derived SCFAs detected in infant feces are acetate and lactate, as well as a small amount of propionate and butyrate [35], in contrast to the adult gut, with a fecal SCFA composition ratio of 3:1:1 for acetate, propionate, and butyrate [36]. The age-related distinction in fecal metabolites could be partially explained by the compositional difference in the gut microbiota. The gut of breast-fed infants is primarily colonized by HMOS-utilizing *Bifidobacterium* spp. And, to a lesser extent, *Bacteroides* spp. [1]. As complementary feeding progresses, the abundances of the butyrate-producing bacteria from the families of Lachnospiraceae and Ruminococcaceae, as well as *Bacteroides* spp., gradually increase with age [2]. Metabolic dependency has been reported between *Bifidobacterium* spp. and butyrate-producing bacteria in utilizing 2′-FL [16]. Here, we demonstrated that *Bacteroides* spp. could also fuel the butyrogenic trophic chain in the presence early-life carbohydrates. *A. caccae* was not able to metabolize either lactose or HMOSs but was dependent on the intermediates produced by *B. thetaiotaomicron* for growth. *A. caccae* could scavenge free monosaccharides, i.e., glucose and galactose, liberated by *B. thetaiotaomicron* from carbohydrate catabolism.

*B. thetaiotaomicron* possesses a range of carbohydrate-active enzymes (CAZymes) predicted to degrade HMOSs, including fucosidases, sialidases, β-galactosidases, and β-hexosaminidases (Figure 4). In contrast, *A. caccae* has a limited catabolic capability. The Sus system of *B. thetaiotaomicron* consists of several membrane-bound proteins and lipoproteins involved in substrate binding, degradation, and internalization into the periplasm [8,9,37]. Fucosidases and sialidases, which are required to initiate HMOS degradation, are often organized in a modular manner adjacent to a transcriptional regulator such as the hybrid two-components system (HTCS) and extracytoplasmic function (ECF) σ-factors, collectively known as polysaccharide utilization loci (PUL) (Appendix A). *B. thetaiotaomicron* is predicted to metabolize HMOSs in a two-step manner, consisting of partial extracellular cleavage followed by oligosaccharide internalization and further degradation for cellular metabolism and storage. We found that *B. thetaiotaomicron* effectively utilized most of the low molecular weight HMOSs, as previously reported [38], with a preference for specific HMOS structures, including 2′-FL, 3-FL, 3′-SL, and 6′-SL, shown by noticeable utilization in the *B. thetaiotaomicron* monoculture at 24 h. Interestingly, at a similar timepoint, an increase in the relative abundance of LNT, LNnT, LNFP I, and LNFP V was observed in the *B. thetaiotaomicron* monoculture. This is likely due to the accumulation of the aforementioned structures from the extracellular cleavage of higher molecular weight HMOSs, such as lacto-N-difucohexaose I (LNDFH I). In order to convert LNDFH I to LNFP I, the externalization of an alpha 1-4 linkage-specific fucosidase is necessary. The characterization of the *B. thetaiotaomicron* fucosidases BT_2970 (in PUL 44) and BT_2192 (in PUL 28) showed that these GH29 enzymes are able to act on alpha 1-4 linkage and alpha 1-3 linkage, respectively [39]. Besides, both fucosidases are predicted to be secretory proteins since they possess signal peptides.

In the co-culture with *A. caccae*, the rapid depletion of lactose was coupled with the degradation of the full range of low molecular weight HMOSs at 24 h. The relatively slow metabolism or repressed degradation of HMOSs in the presence of a preferred substrate, in this case lactose, has been noted previously by Pudlo et al. [40]. *B. thetaiotaomicron* was shown to deprioritize mucin glycan metabolism in the presence of competing complex carbohydrates and monosaccharides [40]. This metabolic plasticity of *B. thetaiotaomicron* has also been demonstrated by the alteration of CAZyme gene expression that resulted in a switch of metabolism from milk to plant carbohydrates after weaning in mice [41]. In addition, the complementary prioritization of specific oligosaccharide structures at the strain level has also been reported to account for the collective fitness of *Bacteroides* spp. [42]. The HMOS foraging capability could enhance bacterial fitness and colonization in the infant gut [43]. As such, several HMOS structures, including 2′-FL, LNFP I, and LDFT, were correlated positively to the abundance of *Bacteroides* spp. in breastfed infants [44]. Intriguingly, *B. thetaiotaomicron* could reciprocally affect the gut glycosylation by regulating the production of fucosylated glycans for its competitive advantage [45].

**Figure 4 microorganisms-08-01513-f004:**
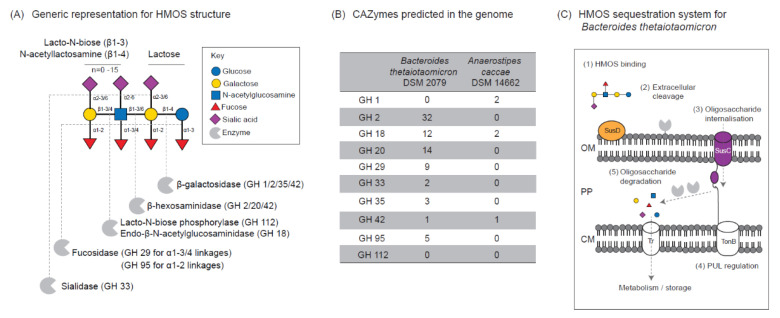
*B. thetaiotaomicron* degraded lactose and HMOSs in the co-culture. (**A**) A generic representation of HMOS structure [46] and prediction of bacterial glycosyl hydrolase (GH) required to cleave the specific linkage. (**B**) Genome prediction for HMOS-degrading carbohydrate-active enzymes (CAZymes) for *B. thetaiotaomicron* and *A. caccae*. (**C**) The proposed mechanism for HMOS sequestration by *B. thetaiotaomicron* using polysaccharide utilization loci (PUL). Abbreviation: HMOS, human milk oligosaccharide; OM, outer membrane; PP, periplasm; CM, cytoplasmic membrane; Tr, transporter; SusC, SusC-like TonB-dependent transporter; SusD, SusD-like outer membrane-binding protein.

Further d- and l-lactate testing showed that the major degraders, i.e., *Bifidobacterium* spp. and *Bacteroides* spp., contributed differently to the lactate isomer pool in the gut (Appendix A). A consistent trait was observed in which *Bacteroides* spp. specifically produced d-lactate whereas *Bifidobacterium* spp. produced only l-lactate when grown in the early-life carbohydrates. The bacterial production of lactate from pyruvate involved the catalysis of stereospecific lactate dehydrogenase. The genome of *B. thetaiotaomicron* encoded only for a d-lactate dehydrogenase (BT_1575), leading to d-lactate production. The lactate-utilizing butyrate-producing bacteria (LUB) could subsequently convert acetate and lactate into butyrate. Our model organism, *A. caccae*, was reported to metabolize both d- and l-lactate [12]. Despite the phenotypic observation, *A. caccae* possessed two l-lactate dehydrogenases (ANACAC_01148 and ANACAC_03769) while no d-lactate dehydrogenase or lactate racemase were found in the genome. The homologous protein check against d-lactate dehydrogenase from *Lactobacillus helveticus* (UniProt identifier P30901) showed no match in the *A. caccae* genome except the general NAD-binding domain. Besides, no domain matching the lactate racemase from *Lactobacillus plantarum* (UniProt identifier F9USS9) was found in the *A. caccae* genome. Nevertheless, as d-lactate conversion was required for the accumulation of butyrate up to 12 mM in the co-cultures, a novel gene/genes could be involved. Other LUB, such as *Eubacterium hallii*, were reported to metabolize both d- and l-lactate, whereas *Roseburia intestinalis*, *Eubacterium rectale*, and *Faecalibacterium prausnitzii* can only metabolize the D form [12]. Ecologically, the lactate isomer pool produced by the degrader community could directly affect substrate availability for LUB. Furthermore, this could potentially incur physiological effects in the host. As the tolerance for d-lactate is lower compared to l-lactate due to the lack of d-lactate dehydrogenase in the human gut, the accumulation of d-lactate is associated with a higher susceptibility to acidosis [47]. The balance of d- and L-lactate was also linked to the risk for d-encephalopathy in patients with short bowel syndrome [48].

## 5. Conclusions

The degrader community could drive the establishment of the microbial network in the infant gut. This forms the basis for the sequential colonization of adult gut-like functional groups, including the lactate-utilizing and butyrate-producing bacteria, followed by the hydrogen-utilizing community (i.e., sulfur-reducing bacteria, reductive acetogens, and methanogens). We showed for the first time that *Bacteroides* spp. Enabled the formation of a butyrogenic trophic chain in the presence of early-life carbohydrates, including lactose and HMOSs. Besides, we revealed the distinct lactate isomer production by *Bifidobacterium* spp. And *Bacteroides* spp., suggesting that the balance of d- and l-lactate could affect gut bacterial structure via cross-feeding.

## Figures and Tables

**Figure 1 microorganisms-08-01513-f001:**
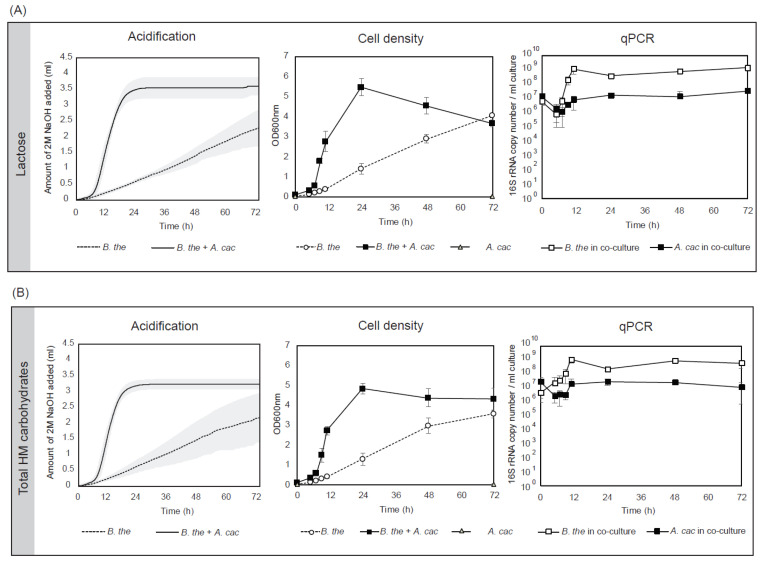
*B. thetaiotaomicron* supported the growth of *A. caccae* in the early-life carbohydrates. The cell density (OD600 nm), acidification, and microbial composition (qPCR) of *B. thetaiotaomicron* monocultures and co-cultures with *A. caccae* in (**A**) lactose and (**B**) total human milk (HM) carbohydrates. Fermentation was performed in an anaerobic bioreactor at pH 6.5. Error bars represent the standard deviation for biological triplicates. No growth was observed for *A. caccae* in the identical medium cultured in anaerobic tubes (Appendix A).

**Figure 2 microorganisms-08-01513-f002:**
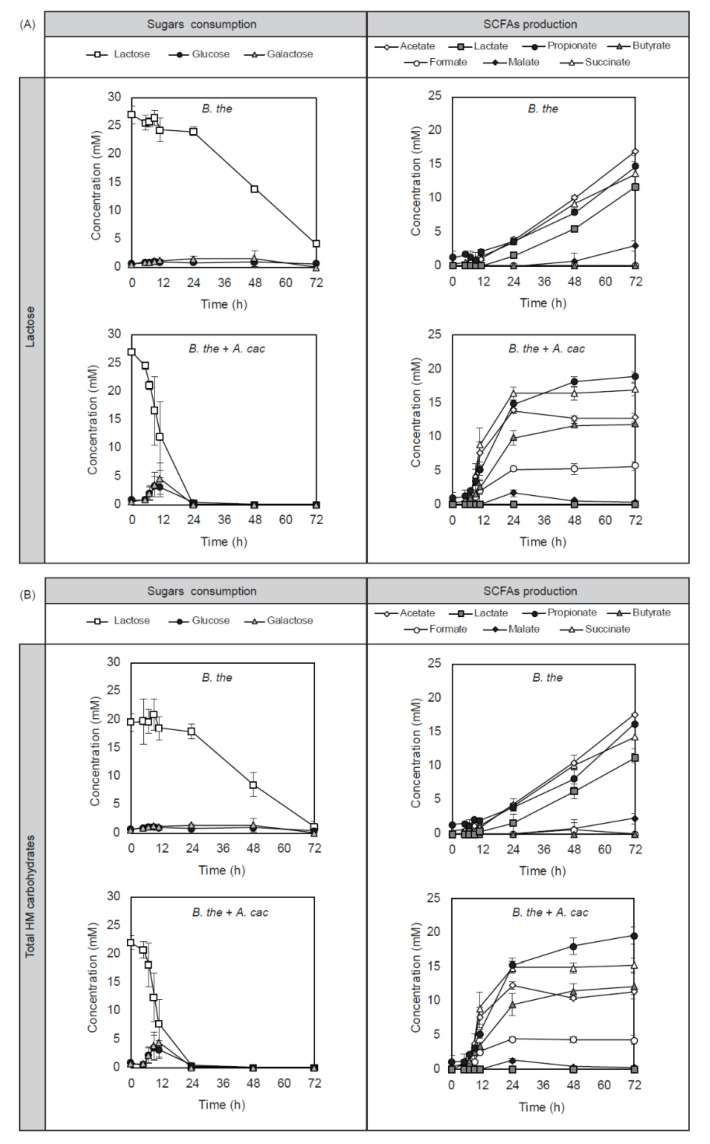
*B. thetaiotaomicron* supported butyrate production of *A. caccae* in the early-life carbohydrates. The sugar and short-chain fatty acid (SCFA) composition of *B. thetaiotaomicron* monocultures and co-cultures with *A. caccae* in basal medium containing (**A**) lactose or (**B**) total HM carbohydrates. Error bars represent the standard deviation for biological triplicates. No metabolite production was detected for *A. caccae* monoculture in the identical medium (Appendix A).

**Figure 3 microorganisms-08-01513-f003:**
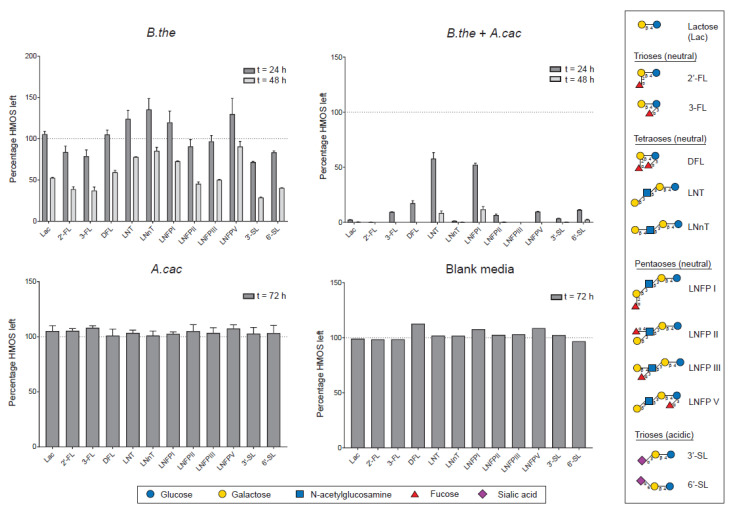
*B. thetaiotaomicron* monoculture and *B. thetaiotaomicron* + *A. caccae* co-culture showed differential utilization of human milk oligosaccharide (HMOS) structures. The percentage of HMOSs left in *B. thetaiotaomicron* monoculture is significantly higher (*p* < 0.05) than *B. thetaiotaomicron* + *A. caccae* co-culture at 24 and 48 h for all structures. Error bars represent the error propagation for the mean of three (for *A. caccae*) or four (for *B. thetaiotaomicron* and *B. thetaiotaomicron* + *A. caccae*) biological replicates measured in technical triplicates. The blank control was measured only in technical triplicates. Abbreviations: 2′-FL, 2′-fucosyllactose; 3-FL, 3-fucosyllactose; DFL, difucosyllactose; LNT, lacto-N-tetraose; LNnT, lacto-N-neotetraose; LNFP I, lacto-N-fucopentaose I; LNFP II, lacto-N-fucopentaose II; LNFP III, lacto-N-fucopentaose III; LNFP V, lacto-N-fucopentaose V; SL, sialyllactose.

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
