# Peer review of "Bacteroides thetaiotaomicron* Fosters the Growth of Butyrate-Producing *Anaerostipes caccae* in the Presence of Lactose and Total Human Milk Carbohydrates"

_microorganisms, 2020, doi:10.3390/microorganisms8101513_

Round 1

Reviewer 1 Report

The manuscript by Chia et al. titled “Bacteroides thetaiotaomicron fosters the growth of butyrate-producing Anaerostipes caccae in the presence of early life carbohydrates” assesses an interesting case of interaction between microorganisms. The techniques used and the methods are adequate.

Comments:

  • In my opinion, the choice of the organism A. caccae is not very clear. A bacterium more characteristic of the early life microbiota could have been chosen. Explain the reasons better.
  • In my view, it is not necessary to abbreviate the species names to B.the and A.cac.
  • In figure 3 report the concentration of HMOs at T0.
  • The discussion section of the study is too long and need to be shortened. Consider separating the conclusions

Author Response

Comments and Suggestions for Authors:

The manuscript by Chia et al. titled “Bacteroides thetaiotaomicron fosters the growth of butyrate-producing Anaerostipes caccae in the presence of early life carbohydrates” assesses an interesting case of interaction between microorganisms. The techniques used and the methods are adequate.

Point 1: In my opinion, the choice of the organism A. caccae is not very clear. A bacterium more characteristic of the early life microbiota could have been chosen. Explain the reasons better.

Response 1: A. caccae is observed in the infant gut microbiota (Backhed et al., 2015; Yatsunenko et al., 2012) and is identified as a key bacterium in regulating food allergy early in life (Feehley et al., 2019). This sentence is added in the introduction section of the manuscript (Line 68-69).

Additional references:

Yatsunenko, T., Rey, F.E., Manary, M.J., Trehan, I., Dominguez-Bello, M.G., Contreras, M., Magris, M., Hidalgo, G., Baldassano, R.N., Anokhin, A.P., Heath, A.C., Warner, B., Reeder, J., Kuczynski, J., Caporaso, J.G., Lozupone, C.A., Lauber, C., Clemente, J.C., Knights, D., Knight, R. and Gordon, J.I. Human gut microbiome viewed across age and geography. Nature 2012, 486: 222-227.

Feehley, T., Plunkett, C., Bao, R., Hong, S., Culleen, E., Belda-Ferre, P., Campbell, E., Aitoro, R., Nocerino, R., Paparo, L., Andrade, J., Antonopoulos, D., Berni Canani, R. and Nagler, C. Healthy infants harbor intestinal bacteria that protect against food allergy. Nature Medicine 2019, 25.

Point 2: In my view, it is not necessary to abbreviate the species names to B.the and A.cac.

Response 2: Revised as suggested throughout the manuscript.

Point 3: In figure 3 report the concentration of HMOs at T0.

Response 3: Figure 3 presented the percentage of HMOS left for B. thetaiotaomicron monoculture and co-culture with A. caccae. For example, the percentage of HMOS left at 24h was calculated by taking the difference between the relative abundance at 0h and 24h, divided by the relative abundance at 0h followed by multiplication with 100%.

Point 4: The discussion section of the study is too long and need to be shortened. Consider separating the conclusions.

Response 4: Revised as suggested with the addition of section 5 for conclusions.

Reviewer 2 Report

The authors conducted lactose and HMO cross-feeding experiment between bacteorides thetaiomicron and anaerostipes caccae.

The experiment was a comparison of mono and co-culture fed on substrates. They compared cell density, abundance of each strain, amount of substrates in the cultures.

They demonstrated that B. thetaiotaomicron is needed for Anaerostipes caccae to produce butyrate in early life.

It would have been nicer if the authors included RT real-time PCR analysis targeting cazymes to prove their theory described in the discussion section. I don't think they can do this now, so please include this in the "further study" or "limitation of this study" paragraph.

Are there any cases that infants having little Bacteroides ended up being sick (diabetes, obesity, autism, atopy, asthma etc.)? This would back up the authors' results better.

Specific comments are below:

The term "early life carbohydrates" is not specific. I suggest "lactose and total human milk carbohydrates"

Asbtract:
Bacteria name should not be shortened this way. Please use B. thetaiotaomicron. Same for other bacteria names.

L27: Bacteroides spp. should be B. thetaiotaomicron because other Bacteroides spp may not have this ability.

Introduction
L43: Bacteroidetes or Bacteroides? L44 says "this genus"

L57: abundant in HMOS

L63: Among the many known butyrogenic bacteria, why did the authors select Anaerostipes caccae? Please explain

Materials and Methods
L82: This reference was published in 1993, suggesting that human milk samples were collected more than 17 years ago?

L170: 501bp is too long for a qPCR. Please check
L172: Not sure what this standard DNA is for. Please explain.

Results

L194: why did they decrease during the first 5h?

Discussion

L281: utilized

Author Response

Comments and Suggestions for Authors:

The authors conducted lactose and HMO cross-feeding experiment between Bacteroides thetaiotaomicron and Anaerostipes caccae.

The experiment was a comparison of mono and co-culture fed on substrates. They compared cell density, abundance of each strain, amount of substrates in the cultures.

They demonstrated that B. thetaiotaomicron is needed for Anaerostipes caccae to produce butyrate in early life.

Point 1: It would have been nicer if the authors included RT real-time PCR analysis targeting cazymes to prove their theory described in the discussion section. I don't think they can do this now, so please include this in the "further study" or "limitation of this study" paragraph.

Response 1: We thank reviewer for the suggestion. qPCR quantification on the CAZymes will provide limited insight on the co-culture system with well sequenced bacterial genomes. For future study, it will be interesting to express the predicted CAZymes for further characterization of the enzymes.

Point 2: Are there any cases that infants having little Bacteroides ended up being sick (diabetes, obesity, autism, atopy, asthma etc.)? This would back up the authors' results better.

Response 2: Infants delivered by caesarean section are observed to be deprived of Bacteroides spp. in the first days of life (Backhed et al., 2015; Martin et al., 2016). To our knowledge, there is no association study suggesting a lower abundance of Bacteroides spp. is linked to a compromised health outcome. However, we demonstrated that B. thetaiotaomicron could support the growth of putative beneficial bacteria, such as A. caccae. A. caccae is observed in the infant gut microbiota (Backhed et al., 2015; Yatsunenko et al., 2012) and is identified as a key bacterium in regulating food allergy early in life (Feehley et al., 2019). Additional sentences in the introduction section of the manuscript (Line 68-69; 72-73).

Additional references:

Yatsunenko, T., Rey, F.E., Manary, M.J., Trehan, I., Dominguez-Bello, M.G., Contreras, M., Magris, M., Hidalgo, G., Baldassano, R.N., Anokhin, A.P., Heath, A.C., Warner, B., Reeder, J., Kuczynski, J., Caporaso, J.G., Lozupone, C.A., Lauber, C., Clemente, J.C., Knights, D., Knight, R. and Gordon, J.I. Human gut microbiome viewed across age and geography. Nature 2012, 486: 222-227.

Feehley, T., Plunkett, C., Bao, R., Hong, S., Culleen, E., Belda-Ferre, P., Campbell, E., Aitoro, R., Nocerino, R., Paparo, L., Andrade, J., Antonopoulos, D., Berni Canani, R. and Nagler, C. Healthy infants harbor intestinal bacteria that protect against food allergy. Nature Medicine 2019, 25.

Point 3: The term "early life carbohydrates" is not specific. I suggest "lactose and total human milk carbohydrates".

Response 3: The titles is revised to ‘Bacteroides thetaiotaomicron fosters the growth of butyrate-producing Anaerostipes caccae in the presence of lactose and total human milk carbohydrates’.

Point 4: Bacteria name should not be shortened this way. Please use B. thetaiotaomicron. Same for other bacteria names.

Response 4: Revised as suggested throughout the manuscript.

Point 5: L27: Bacteroides spp. should be B. thetaiotaomicron because other Bacteroides spp may not have this ability.

Response 5: Revised as suggested (Line 30).

Point 6: Introduction L43: Bacteroidetes or Bacteroides? L44 says "this genus"

Response 6: This genus refers to Bacteroides. The sentence is revised to ‘infants delivered by caesarean section are deprived of Bacteroides spp. in the first days of life’ (Line 45).

Point 7: L57: abundant in HMOS

Response 7: This sentence specifies that 2’-FL is an abundant HMOS.

Point 8: L63: Among the many known butyrogenic bacteria, why did the authors select Anaerostipes caccae? Please explain

Response 8: A. caccae is observed in the infant gut microbiota (Backhed et al., 2015; Yatsunenko et al., 2012) and is identified as a key bacterium in regulating food allergy early in life (Feehley et al., 2019). This sentence is added in the introduction section of the manuscript (Line 68-69).

Additional references:

Yatsunenko, T., Rey, F.E., Manary, M.J., Trehan, I., Dominguez-Bello, M.G., Contreras, M., Magris, M., Hidalgo, G., Baldassano, R.N., Anokhin, A.P., Heath, A.C., Warner, B., Reeder, J., Kuczynski, J., Caporaso, J.G., Lozupone, C.A., Lauber, C., Clemente, J.C., Knights, D., Knight, R. and Gordon, J.I. Human gut microbiome viewed across age and geography. Nature 2012, 486: 222-227.

Feehley, T., Plunkett, C., Bao, R., Hong, S., Culleen, E., Belda-Ferre, P., Campbell, E., Aitoro, R., Nocerino, R., Paparo, L., Andrade, J., Antonopoulos, D., Berni Canani, R. and Nagler, C. Healthy infants harbor intestinal bacteria that protect against food allergy. Nature Medicine 2019, 25.

Point 9: Materials and Methods L82: This reference was published in 1993, suggesting that human milk samples were collected more than 17 years ago?

Response 9: The total human milk carbohydrates were prepared from human milk samples pooled in the 1990s, as stated in the publication by Thurl et al. (1993) (Line 85-88). Composition of the freeze-dried total HM carbohydrates is rather stable over time, with the GPC-RI analysis presented in Chia et al. (2020).

Point 10: L170: 501bp is too long for a qPCR. Please check

Response 10: The PCR product length provided is correct.

Point 11: L172: Not sure what this standard DNA is for. Please explain.

Response 11: A longer standard template DNA was used in order to ensure proper annealing of the primers. For example, we used A. caccae standard template of around 1.4 kbp, synthesized using primer pair of 27F (5’-AGAGTTTGATCCTGGCTCAG-3’ and 1492R (5'-GGTTACCTTGTTACGACTT-3'). qPCR was performed using A. caccae 16s specific primer pair OFF2555 5'-GCGTAGGTGGCATGGTAAGT-3'; OFF2556 5'-CTGCACTCCAGCATGACAGT-3' (Veiga et al. 2010) with 83 bp product. As such, the DNA template is longer than the product range and this could avoid low annealing efficiency due to truncated template.

Point 12: Results L194: why did they decrease during the first 5h?

Response 12: Around 6 log of cells were inoculated for B. thetaiotaomicron and A. caccae and both of the strains decreased 10-fold in abundance at the first 5 h. The initial decline is due the lag phase.

Point 13: Discussion L281: utilized

Response 13: UK English is used throughout the manuscript.

Reviewer 3 Report

This study investigated the cross-feeding interaction between B.the and A.cac in presence of human milk carbohyrates using bioreactor mono/co-culture system. The methods and main minding is straight-forward and intriguing, however, there are several questions need to be answered.

  1. Why did authors select A. caccae to investigate the cross-feeding ability of B. thetaiotaomicron? There are several references dealing with the nourishing effect of B.the on other butyrate-producing bacteria (e.g. Faecalibacterium sp.) or most abundant infant microbiota (e.g. Bifidobacterium spp.). In infant gut, A. caccae is not an abundant bacteria. Is there any importance to enhance the growth of this bacteria in an aspect of promoting infants' health? How do you think the potential harmful effects of increasing this specific species in infants' gut or the safety assessment of this species compared to other butyrate-producing bacterial species?
  2.  Is the cross-feeding effect of B.the on A.cac is specifically dependent on the utilization of HMOS? Did you conduct the cross-feeding experiment using other carbohydrate sources, e.g. starch or non-digestible fiber, etc.? If not, the expecting results should be described in Discussion based on reference and genome analysis.
  3. Statistical significance should be tested and presented in all Figures, and the results must be interpreted based on the statistical significance. 

Author Response

Comments and Suggestions for Authors:

This study investigated the cross-feeding interaction between B.the and A.cac in presence of human milk carbohydrates using bioreactor mono/co-culture system. The methods and main finding are straight-forward and intriguing, however, there are several questions need to be answered.

Point 1: Why did authors select A. caccae to investigate the cross-feeding ability of B. thetaiotaomicron? There are several references dealing with the nourishing effect of B.the on other butyrate-producing bacteria (e.g. Faecalibacterium sp.) or most abundant infant microbiota (e.g. Bifidobacterium spp.). In infant gut, A. caccae is not an abundant bacterium. Is there any importance to enhance the growth of this bacteria in an aspect of promoting infants' health? How do you think the potential harmful effects of increasing this specific species in infants' gut or the safety assessment of this species compared to other butyrate-producing bacterial species?

Response 1: A. caccae is used as a representative species from the Lachnospiraceae family, to demonstrate the proof of concept that B. thetaiotaomicron could support the growth of butyrate-producing bacteria. A. caccae is observed in the infant gut microbiota (Backhed et al., 2015; Yatsunenko et al., 2012) and is identified as a key bacterium in regulating food allergy early in life (Feehley et al., 2019). Additional sentences in the introduction section of the manuscript (Line 68-69; 72-73).

Additional references:

Yatsunenko, T., Rey, F.E., Manary, M.J., Trehan, I., Dominguez-Bello, M.G., Contreras, M., Magris, M., Hidalgo, G., Baldassano, R.N., Anokhin, A.P., Heath, A.C., Warner, B., Reeder, J., Kuczynski, J., Caporaso, J.G., Lozupone, C.A., Lauber, C., Clemente, J.C., Knights, D., Knight, R. and Gordon, J.I. Human gut microbiome viewed across age and geography. Nature 2012, 486: 222-227.

Feehley, T., Plunkett, C., Bao, R., Hong, S., Culleen, E., Belda-Ferre, P., Campbell, E., Aitoro, R., Nocerino, R., Paparo, L., Andrade, J., Antonopoulos, D., Berni Canani, R. and Nagler, C. Healthy infants harbor intestinal bacteria that protect against food allergy. Nature Medicine 2019, 25.

Point 2: Is the cross-feeding effect of B.the on A.cac is specifically dependent on the utilization of HMOS? Did you conduct the cross-feeding experiment using other carbohydrate sources, e.g. starch or non-digestible fiber, etc.? If not, the expecting results should be described in Discussion based on reference and genome analysis.

Response 2: As this study is focusing on the early life gut environment, in which breastmilk is the sole nutritional source for most infants, the utilization of HMOS is investigated. Other carbohydrate sources are interesting for later stages of life, in which several papers described B. thetaiotaomicron utilization of starch (Rodriguez-Castaño et al., 2019), xylan (Zhang et al., 2014) and arabinogalactan (Cartmell et al., 2018).  

Interesting further reading:

Rodriguez-Castaño, G.P., Dorris, M.R., Liu, X., Bolling, B.W., Acosta-Gonzalez, A., Rey, F.E. Bacteroides thetaiotaomicron Starch Utilization Promotes Quercetin Degradation and Butyrate Production by Eubacterium ramulus. Front. Microbiol 2019, 10: 1145.

Cartmell, A., Muñoz-Muñoz, J., Briggs, J.A., Ndeh, D.A., Lowe, E.C., Baslé, A., Terrapon, N., Stott, K., Heunis, T., Gray J., Yu, L., Dupree, P., Fernandes, P.Z., Shah, S., Williams, S.J., Labourel, A., Trost, M., Henrissat, B., Gilbert, H.J. A surface endogalactanase in Bacteroides thetaiotaomicron confers keystone status for arabinogalactan degradation. Nature Microbiology 2018, 3: 1314-1326.

Zhang, M., Chekan, J.R., Dodd, D., Hong, P-Y., Radlinski, L., Revindran, V., Nair, S.K., Mackie, R.I., Cann, I. Xylan utilization in human gut commensal bacteria is orchestrated by unique modular organization of polysaccharide-degrading enzymes. PNAS 2014, 111(35): E3708-E3717.

Point 3: Statistical significance should be tested and presented in all Figures, and the results must be interpreted based on the statistical significance.

Response 3: Statistical analysis was performed on Figure 3, with additional finding added to the figure caption (Line 262-264). The percentage of HMOS left in B. thetaiotaomicron monoculture is significantly higher (p < 0.05) than B. thetaiotaomicron + A. caccae co-culture at 24 and 48 h for all structures. Statistics were performed using t-test and corrected for multiple testing using FDR correction for multiple comparisons. P-values < 0.05 were considered significant (additional description for statistical analysis in Material & Methods, Line 190-192)

Round 2

Reviewer 3 Report

The statistical difference should be represented using an asterisk in figures.